



# SynRad v1.0: A radar forward operator to generate synthetic radar return signals from volcanic ash clouds.

Vishnu Nair[1], Anujah Mohanathan[2], Michael Herzog[1], David G. Macfarlane[3], and Duncan A. Robertson[3]

[1]Department of Geography, University of Cambridge, Cambridge, UK.
[2]Department of Physics, University of Cambridge, Cambridge, UK.
[3]SUPA School of Physics and Astronomy, University of St Andrews, UK.

**Correspondence:** Vishnu Nair (vn294@cam.ac.uk)

**Abstract.** In this work, SynRad, a new radar forward operator for the ATHAM volcanic plume model is introduced. The operator is designed to generate synthetic radar signals from ground-based radars for volcanic ash clouds simulated by ATHAM. A key novelty of SynRad is a ray tracing module which traces radar beams from the antenna to the ash cloud and calculates path attenuation due to hydrometeors and ash. The operator is designed to be compatible with the one-moment microphysics scheme in ATHAM, but can be easily extended to other one- or two-moment schemes in ATHAM or any weather prediction model. The operator can be used to test candidate locations at which to operationally deploy portable high frequency or multi-frequency (from long to short wavelength) radar(s). Optimal frequency or frequencies (for a multi-frequency radar) can be identified which balances the trade-off between a higher return signal and the higher path attenuation that comes at these higher frequencies. A case study of the eruption of the Raikoke volcano in 2019 is used to evaluate the performance of SynRad. The measurement process of a C-band radar is simulated using SynRad and the operator was able to generate realistic fields of the equivalent radar reflectivities, echotops and vertical maximum intensities. Ideally, higher frequency microwave radars will be designed and constructed specifically for monitoring volcanic eruptions. This is certainly possible in the coming years which makes feasibility studies on the capability of higher frequency radars timely.

## 1 Introduction

Active monitoring of volcanic eruptions and the subsequent ash cloud dispersal is an area of significant interest due to their human, social and economical effects. These include direct impacts such as the hazard to aviation via the tephra injected into the atmosphere, and the primary and secondary impacts of the volcanic ash accumulating on the ground causing damage to life and property in the surrounding areas (Jenkins et al., 2015). Real-time monitoring of the volcanic ash cloud distribution and dispersal have been done mainly using satellite measurements, albeit with comparatively coarse spatial (order of a few kilometers) and temporal (15-30 minutes) resolutions. Timely information and predictions of the ash cloud dispersal are vital as most encounters with the ash cloud have historically happened within minutes to a few hours.

Ground-based microwave radars present a unique opportunity to monitor ash clouds with relatively higher spatial (less than a few hundred meters) and temporal resolutions (every few minutes), and for 24 hours a day. These radars measure the back-scattered energy returned to the radar dish as a result of interactions between the pulse (or continuous wave) of



electromagnetic energy and the scatterers. The scatterers can be volcanic ash particles, hydrometeors such as cloud, ice or precipitation, or aerosol particles. These interactions depend on the size, shape, orientation and distribution of the particles, and the frequency and polarization of the electromagnetic radiation. The back-scattered signals measured at the radar receiving antenna is then typically converted to equivalent radar reflectivities (in decibels) which is the most common and familiar quantity in meteorology.

Permanent monitoring of volcanic ash clouds using weather radars operating at lower frequencies, i.e. S-, C- and X-bands has become quite common, buoyed by the benefits of continuous quantitative retrieval of two key source parameters that are fed into long-range ash dispersion models - the eruption plume height and the tephra eruption rate and mass. There are several ground-based weather networks operational in Alaska, Iceland and Italy (to name a few) which have been key in monitoring eruptions. Notable examples include the monitoring using C-band radars of the eruption of the Hekla volcano in 2000 (Lacasse

et al., 2004), the eruption of the Grímsvötn volcano in Iceland in 2004 (Marzano et al., 2010a), the Augustine volcanic eruption in 2006 in Alaska using S-band weather radar imagery (Marzano et al., 2010b), the eruption of the Redoubt volcano in 2009 (Schneider and Hoblitt, 2013) and the Eyjafjallajökull eruption in 2010 using C-band weather radars (Marzano et al., 2011), and the Mt. Etna eruption in 2011 in Italy using a mobile X-band dual-polarization radar (Marzano et al., 2013).

The ready-availability of lower frequency weather radars make them convenient to monitor ash clouds (Marzano et al.,

2013); however, particle size distributions and refractive indices of ash particles are quite different from that of cloud droplets, ice crystals, rain drops, graupel or hail. The refractive index of ash is generally lower than that of water, and the ash particle sizes (especially far away from the vent) are smaller than raindrops which means that frequencies higher than those used in weather radars have possible benefits while monitoring and measuring volcanic ash concentrations. Additionally, successfully detecting a volcanic ash cloud using ground-based radar systems depends not only on the radar system characteristics, but also

on the radar range, i.e., the location where the radar is operationally deployed with respect to the vent and the ash cloud. These factors can be combined into a single parameter - the minimum detectable signal (MDS); if the return signal from a radar sample volume is higher than the MDS, this means that the sample will be detected by the radar. Increasing the operational frequency or reducing the range can result in a stronger return signal from the scattering volume. However, at higher frequencies (millimeter wavelengths), the two-way path attenuation plays a key role and determines the degree to which the transmitted signal can

penetrate into the ash cloud. Therefore, it is important to identify plausible locations relative to the vent and the dispersing ash cloud at which to operationally deploy these millimeter-wave radars (which are often portable) which allows the radar to penetrate and scan the full extent of the ash cloud in three dimensions. This analysis can be done by using 'radar forward operators' which are numerical operators that can generate synthetic radar signals from gridded data or three-dimensional output from numerical simulations of volcanic eruptions.

Given certain meteorological conditions, radar forward operators have been used as a key tool to derive synthetic radar observations from numerical weather prediction (NWP) model prognostic variables. They also form a key part of inversion algorithms used to retrieve radar observable variables or in data assimilation. These operators include numerical descriptions of the different physical aspects of radar measurements. Modeled clouds are converted to equivalent radar reflectivities in order to compare them with actual radar observations (from either ground-based radars or radars on satellites). Most operators





developed so far have been polarimetric radar operators used for hydrometeor classification and improvement in weather radar data quality and rainfall measurements in addition to the validation (of microphysics modules) and assimilation of radar reflectivities into NWP models (Pfeifer et al., 2008; Cheong et al., 2008; Jung et al., 2008; Ryzhkov et al., 2011; Augros et al., 2016; Zeng et al., 2016; Oue et al., 2020; Shrestha et al., 2022; Li et al., 2022). In this paper, we introduce SynRad, a new radar forward operator which has been exclusively designed to synthetically calculate the return signals from volcanic ash clouds

and hydrometeors. Even though conventional forward operators calculate the attenuated or corrected reflectivity, they do not trace the radar beam and its extinction through the cloud under study. A key aspect of SynRad is a ray tracing module that calculates the extinction of the transmitted (and reflected) radar signal due to interactions within the ash cloud. As previously discussed, the advent of portable millimeter wavelength radar systems has made it important to include path attenuation in all relevant studies. A forward radar operator designed specifically for volcanic ash cloud tracking then plays a significant role

in such cases, allowing for a purpose quite different from its weather counterparts by synthetically calculating the attenuated return signal at the radar.

Just as forward operators can help to improve numerical models, conversely they have the potential to complement the process of designing and operationally deploying portable or fixed radars. With millimeter wave radars being considered specifically for monitoring volcanic ash clouds (Macfarlane et al., 2021), the development of such an operator and a methodology is

timely to inform radar design and the overall suitability of such radars to detect volcanic ash clouds, and characterizing the sizes and concentrations of the volcanic ash (with multiple-frequency radars), immediately post-eruption. Key to this is identifying the optimal location and range from the volcano vent where it would best serve its purpose. For a known set of radar system characteristics, the synthetic radar observables can then help decide optimum locations for radar deployment. It can also be used to evaluate the performance of existing fixed weather radars (with a certain MDS) for volcanic ash cloud monitoring. This

will be the focus of this paper as we introduce the work flow of SynRad and evaluate the performance of a C-band weather radar to detect the ash cloud from the 2018 eruption of the Raikoke volcano. This is also the first time the said eruption has been numerically modeled and the results presented.

In section 2, the type of model data that serves as the input to SynRad is described. In section 3, a detailed workflow of SynRad, including the theory and computational modular structure is introduced. In section 4, the main results and performance

of SynRad in the C-band are evaluated by using the operator on numerical simulation data of the 2019 eruption of the Raikoke volcano. Finally, in section 5, the conclusions are presented along with future plans for SynRad.

## 2   Description of the data - input to SynRad

SynRad v1.0 generates synthetic radar signals from three-dimensional output generated by the Active Tracer High-Resolution Atmospheric Model (ATHAM). ATHAM is a three-dimensional, non-hydrostatic atmospheric model that has been specifically

designed to simulate characteristics of volcanic eruption plumes (Oberhuber et al., 1998; Herzog et al., 2003; Herzog and Graf, 2010). ATHAM predicts the behaviour of a multi-component system consisting of a gas-particle mixture with arbitrary tracer concentrations. Unique features of ATHAM are its dynamically and thermodynamically active tracers, as well as its





ability to represent strongly divergent flows with large vertical accelerations. The vent size, exit velocity, temperature and composition of the mixture are prescribed as a function of time. A single-moment cloud microphysical module (Herzog et al., 1998) predicts the presence of condensed water, ice, and graupel, and the effect of phase changes on the plume's heat budget but does not currently include interactions of hydrometeors with ash particles. Typical three-dimensional output generated by ATHAM includes netCDF files for every minute for the wind velocities, pressure, mixture potential and in-situ temperatures and densities, specific concentrations of the different ash categories, hydrometeor categories, water vapor and gases like $SO_2$.

## 3 SynRad methodology, workflow and assumptions

SynRad v1.0 generates synthetic radar signals by calculating the return power, path attenuation and the effective radar reflectivity (a common radar observable) from ATHAM three-dimensional output and user-specified radar characteristics, ie, the radar position, wavelength (or frequency), transmit power, the one-way 3 dB antenna beamwidth and the range bin width (or resolution). For a pulsed radar this is the pulse width, whereas for a frequency modulated continuous wave (FMCW) radar this depends on the signal bandwidth. The standard workflow of SynRad is shown in figure 1 and detailed in the following sub-sections. For a given radar location, rays are traced from the antenna to each ATHAM grid cell centre (which acts as the radar sample volume) and path attenuation is calculated at different points along this ray. In other words, a radar is placed in the numerical domain and the beam paths are traced over the three-dimensional output, cell-by-cell. Calculations are performed on each cell in the grid, i.e. no volume scan measurements are simulated. In this section, the methodology behind SynRad is introduced. This includes the theory behind the physical aspects of radar measurement, and how this theory is implemented in SynRad via the different modules. But first, we start with the assumptions made during synthetic signal generation.

### 3.1 Assumptions made in SynRad

In this sub-section, we list the assumptions made in the current version of SynRad.

1. All scattering particles (ash and hydrometeors) are homogeneous dielectric spheres that are assumed to occupy the entire resolution volume.

2. The atmosphere in which the synthetic radar beam is propagating in the model domain is assumed to be non-attenuating, i.e. we neglect attenuation due to atmospheric gases (oxygen, water vapor and nitrogen) and only consider attenuation due to condensed water, ice crystals and graupel as these would dominate the attenuation field. Attenuation due to volcanic $SO_2$ is also neglected. The dominating attenuation in this case is due to the ash particles and hydrometeors and hence the decision was made to neglect that due to gases.

3. Beam bending due to atmospheric refraction along its trajectory is neglected.

4. The effects of multiple scattering within a resolution volume is ignored.





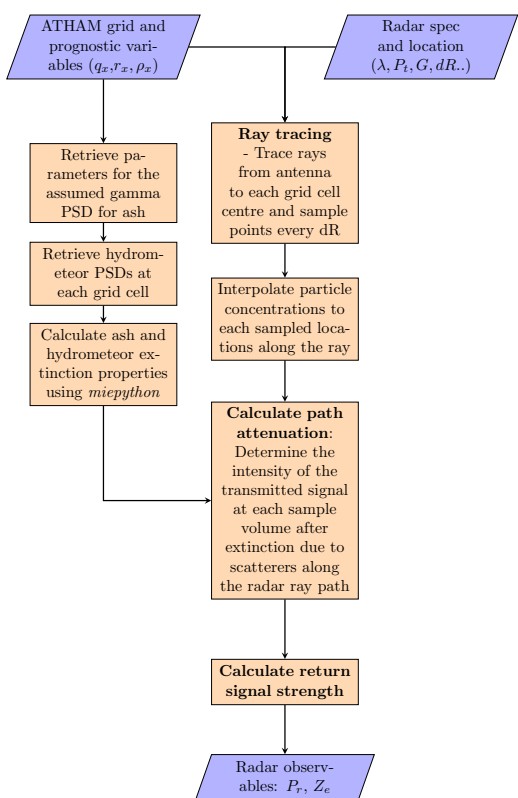

**Figure 1.** SynRad workflow.

5. A single polarized radar is considered where the incident and back-scattered waves are linearly polarized. A polari-
   metric radar sends out signals that are oriented vertically and horizontally and by comparing the reflected signals from
   both, different precipitation types can be identified. Here, we only consider horizontally polarized signals. The effect of
   polarization is hence not considered but this is something that will be added in upcoming versions.

## 3.2 Radar measurement theory

For a radar with transmitted power $P_t$ (W), antenna gain $G$, one-way normalized radiation pattern $|f(\theta_1, \phi_1)|^2$ (where $(\theta, \phi)$
defines the half-power beam-width (for one-way transmission)), and effective antenna aperture $A_e$, the power received by the
antenna from all volume-distributed targets in a radar resolution volume $dV$ that is spanned by a solid angle $d\Omega$, and range
resolution $dR$ is given by (Probert-Jones, 1962)

$$\mathrm{d}P_r = \frac{P_t d\Omega}{(4\pi)} G|f(\theta,\phi)|^2 \, dR \cdot A_e|f(\theta,\phi)|^2 \frac{1}{4\pi R^2} \cdot \eta \cdot d\Omega, \tag{1}$$

where $\eta$ is the volumetric radar reflectivity (m$^{-1}$) and $\mathrm{d}\Omega$ is subtended by an area $\mathrm{d}a$ (m$^2$) at a range $R$ (m). We can relate
$A_e$ to the antenna gain by $A_e = \frac{\lambda^2}{4\pi} G$, where $\lambda$ is the radar wavelength (m). The total power received at the same instance of





time is then obtained by integrating in range over the resolution $\delta$ and over the solid angle $\Omega$ subtended by that portion of the

surface of the sphere (of radius $R$) giving

$$P_r = \int\limits_{R}^{R+\delta} \int\limits_{\Omega} dP_r = \frac{P_t \lambda^2 G^2}{(4\pi)^3} \int\limits_{R}^{R+\delta} \frac{dR}{R^2} \cdot \int\limits_{d\Omega} |f(\theta,\phi)|^4 \cdot \eta, \tag{2}$$

Since $R \gg \delta$, the first integral gives

$$\int\limits_{R}^{R+\delta} \frac{dR}{R^2} = \frac{\delta}{R^2}, \tag{3}$$

and the second integral can be evaluated to (Probert-Jones, 1962)

$$\int\limits_{d\Omega} |f(\theta,\phi)|^4 = \frac{\pi \phi_1 \theta_1}{8 ln 2}. \tag{4}$$

For an FMCW radar, the range resolution $\delta = c/2B$ where $c$ is the speed of light and $B$ is the chirp bandwidth. This finally

yields

$$P_r = P_0 \frac{\lambda^2 G^2}{(4\pi)^3} \frac{c}{2BR^2} \cdot \frac{\pi \phi_1 \theta_1}{8 ln 2} \cdot \eta. \tag{5}$$

This is the form of the radar range equation that forms the basis of SynRad. Equation (5) is valid for a non-attenuating medium.

For an attenuating medium, the radar equation is

$$P_r = P_0 \frac{\lambda^2 G^2}{(4\pi)^3} \frac{c}{2BR^2} \cdot \frac{\pi \phi_1 \theta_1}{8 ln 2} \cdot \eta \cdot L^2, \tag{6}$$

where $L$ is the one-way path attenuation factor from the radar antenna to the considered range bin and quantifies the damping

of the radar signal as it propagates through the atmosphere and the volcanic ash plume. Equation (6) can be simplified to the

working form used in SynRad

$$P_r = C_r \frac{Z}{R^2} L^2 \tag{7}$$

where $C_r$ is the radar constant that is specific to the radar system and $Z$ is the radar reflectivity factor ($\mathrm{m}^6\mathrm{m}^{-3}$). The relation

between $\eta$ and $Z$ is introduced in section 3.3.2.

## 3.3 Synthetic radar signal simulation

The synthetic radar signal simulation model involves two modules: 1) the ray tracing module which simulates the propagation

and extinction of the radar beam by hydrometeors and volcanic ash, and (2) the radar reflectivity module which takes in the

specific concentrations (and number concentrations for a two-moment microphysics scheme) of the scatterers and calculates

the radar reflectivities.





### 3.3.1 Ray tracing and path attenuation calculation

This module measures the extinction of the radar signal due to ash or hydrometeors along its path. The exponential decay of

the amplitude of a radar signal $P$ propagating a radial distance $r$ in an attenuating medium can be calculated by

$$P = P_0 e^{-\int_0^r \kappa(r)dr} \tag{8}$$

where $\kappa$ $(\mathrm{m}^{-1})$ is the volumetric specific attenuation or extinction coefficient given by

$$\kappa = \int_{D_1}^{D_2} \sigma_e(D, f, \epsilon) N(D) dD, \tag{9}$$

where $\sigma_e = \sigma_s + \sigma_a$ is the extinction cross-section $(\mathrm{m}^2)$, and $\sigma_s, \sigma_a$ are the scattering and absorption cross-sections $(\mathrm{m}^2)$ for

a particle of a given diameter $D$ at a frequency $f$ and permittivity $\epsilon$. The exponential in the right-hand side of equation (8) is usually called the one-way path attenuation factor $L$ (which appears in equation (6)), i.e.,

$$L(r) = e^{-\int_0^r \kappa(r)dr}. \tag{10}$$

With the radar location as the starting point, a ray is traced to the centre of each cell in the three-dimensional grid of the host model output. Each ray is then divided into $n$ segments and the values of the ash and hydrometeor specific concentrations

are linearly interpolated to the end-points of each segment. Starting with an initial transmitted power $P_0$ at the radar, the transmittance $P/P_0$ is calculated at each segment centre using equation (8). The final value of $P$ at the cell centre of the segment which acts as the end of the ray is calculated as a cumulative product of the transmittance at each ray segment. In this manner, rays are traced to every cell of the 3-d grid with total scattering specific concentrations above a threshold. Each cell corresponds to a radar resolution volume, which is associated with a particular value of $P_r$ and $Z$ according to equation

(7). For a given set of radar system characteristics with a minimum detectable signal (MDS), we then discard all the rays where the return signal strength $P_r$ is below the MDS. Every microwave radar system is assigned an MDS which assumes the noise figure of the radar receiver and is generally an available characteristic which assumes different values for a specific radar system. Alternately, the MDS can be calculated from (7) by using a known minimum dBZ for a specific radar system.

### 3.3.2 Radar reflectivity module

The radar reflectivity module takes in the host-model prognostic variables such as specific concentration $q$ $(\mathrm{kg\,kg}^{-1})$ (for one-moment microphysics schemes), and total number concentration $N_T$ $(\mathrm{m}^{-3})$ (for two-moment schemes) and returns the equivalent radar reflectivity $Z_e$ (dBZ). $Z_e$ is the radar-observed reflectivity and includes contributions from both ash ($Z_{\mathrm{e,ash}}$) and hydrometeors which include (for ATHAM) cloud droplets ($Z_{\mathrm{e,cld}}$), pure rainwater ($Z_{\mathrm{e,rn}}$), ice crystals ($Z_{\mathrm{e,ice}}$) and graupel ($Z_{\mathrm{e,graup}}$). A separate gamma function is used to describe the size distribution of ash and hydrometeors. Conventional single

polarization Doppler precipitation radars measure the horizontally polarized volumetric radar reflectivity ($\eta$ in equation (6)),



which is the summation over all back-scattering cross-sections in a resolution volume at a particular bin range, as

$$\eta_H = \int_{D_1}^{D_2} \sigma_b(D, f, \epsilon) N(D, f, \epsilon) \mathrm{d}D, \tag{11}$$

where $\sigma_b$ $(\mathrm{m}^2)$ is the back-scattering cross-sectional area, $N(D)$ is the number of scattering particles of diameter $D$ per radar sample volume $(\mathrm{m}^{-4})$, and $(D_1, D_2)$ are the minimum and maximum particle diameters respectively in the sample volume.

Since we only consider single-polarization radars in this study the subscript '$H$' will be dropped for the rest of the manuscript.

The module calculates the back-scattering cross-section and thereafter the radar reflectivity based on both the Rayleigh approximation and Mie theory. The Rayleigh scattering approximation is valid for a spherical particle with diameter $D \ll \lambda$ (or $\frac{2\pi D}{\lambda} \ll 1$) in which case $\sigma_b$ can be approximated as

$$\sigma_b(D) = \frac{\pi^5 |K|^2}{\lambda^4} D^6 \tag{12}$$

where $|K|^2$ is the complex dielectric factor of the scatterer.

Combining equations (11) and (12), under the Rayleigh approximation, gives

$$\eta = \frac{\pi^5 |K|^2}{\lambda^4} \int_{D_1}^{D_2} D^6 N(D) dD, \tag{13}$$

The estimate of the radar reflectivity factor from the droplet size distributions can be expressed as the sixth moment of the drop size distribution (DSD)

$$Z = \int_0^\infty N(D) D^6 dD, \tag{14}$$

for which an analytical solution exists if we are assuming a gamma distribution for the scatterer sizes (details in 3.3.3), giving

$$\eta = \frac{\pi^5 |K|^2}{\lambda^4} Z. \tag{15}$$

Since the Rayleigh approximation does not always hold and the scatterers could be categories other than liquid water (eg. ice, ash, graupel etc), the default is to calculate $\eta$ using the Mie solutions and relate this to an 'effective' or 'equivalent' reflectivity

factor $Z_e$ which is defined as

$$Z_e = \frac{\lambda^4}{\pi^5 |K_w|^2} \eta. \tag{16}$$

Here $|K_w|^2$ is the dielectric factor for water, which is to account for the fact that scatterers are expected to be spherical water droplets. Hence, for water droplets in the Rayleigh regime, $Z_e = Z$. The equivalent reflectivity factor $Z_e$ is therefore the radar reflectivity of a target consisting of water drops (with $D \ll \lambda$) which would produce the same reflectivity as that of a target

with known properties (AMS Glossary of Meteorology).



The Mie solutions for the back-scattering cross-section of $N$ dielectric homogeneous spherical particles are given by (Mie, 1908)

$$\sigma_b = \frac{\lambda^2}{2\pi} \sum_{n=1}^{N} (2n+1)|a_n|^2 + |b_n|^2, \tag{17}$$

where the Mie coefficients $a_n$ and $b_n$ are spherical Bessel functions which depends on $m$ and a size parameter $x = \pi D/\lambda$. These solutions provide the more accurate description but can be computationally intensive. In this module, the python package *miepython* is used to efficiently calculate the back-scattering cross-sections according to Mie theory and following the procedure described by Wiscombe (1979). *miepython* takes as input the size parameter $x$ and the refractive index $m$ of the scatterer. Therefore, while using the Mie formulas, $Z_e$ is calculated using equations (16) with $\eta$ calculated using equation (11) as a function of the frequency $f$, refractive index $m$ of the hydrometeor under consideration, and in which $\sigma_b$ is calculated using *miepython*. Section 3.4 details the calculation of the complex permittivity and the refractive indices for the scatterers to be used in miepython.

It is worth mentioning that the equivalent reflectivity factor $Z_e$ and the received power $P_r$ are also expressed in logarithmic powers dBZ$_e$ and dBm respectively as their values can vary over orders of magnitude. These are expressed as dBZ$_e = \log_{10}(Z_e/Z_0)$ and dBm $= \log_{10}(P_r/P_{r0})$ where $Z_0 = 1\,\mathrm{mm}^6\mathrm{m}^{-3}$ is a reference value calculated for a droplet/particle of size 1mm and $P_{r0}$ is a reference power of 1mW. Another point to note is that since $Z_e$ is calculated in standard units of $\mathrm{m}^6\mathrm{m}^{-3}$, $Z_0$ has to be converted to the same units by multiplying with a factor of $10^{-18}$.

### 3.3.3 Particle size distribution

In this subsection, the different forms of the particle size distributions (PSD) used to describe the particles existing per unit resolution volume and unit size is introduced. The code can work with both a monodisperse and a polydisperse distribution of particles. Furthermore, a gamma function is usually used to describe a polydisperse PSD of ash or hydrometeors. A general form of the gamma distribution function used to describe the polydisperse PSD of any particle category is given by

$$N(D_x) = N_{Tx} \frac{\alpha_x}{\Gamma(\nu_x)} \lambda_x^{\alpha_x \nu_x} D_x^{\alpha_x \nu_x - 1} e^{-(\lambda_x D_x)^{\alpha_x}} \tag{18}$$

where $N(D_x)$ is the total number concentration per unit volume of ash or hydrometeor of category $x$ of diameter $D_x$ (m$^{-4}$), $N_{Tx}$ is the total number concentration of the same category (m$^{-3}$), $\lambda_x$ is the slope parameter (m$^{-1}$), $\nu_x$ and $\alpha_x$ are the dispersion functions (dimensionless) and $\Gamma$ is the gamma function. This *generalized* gamma function can be simplified further by setting $\alpha_x = 1$ to give the *gamma function* which is a three-parameter function involving $N_{0x}$, $\lambda_x$ and $\nu_x$ expressed as

$$N(D_x) = N_{0x} D_x^{\nu_x - 1} e^{-(\lambda_x D_x)}, \tag{19}$$

where $N_{0x}$ is the total number concentration parameter (m$^{-(\nu_x + 3)}$) given by

$$N_{0x} = \frac{N_{Tx}}{\Gamma(\nu_x)} \lambda_x^{\nu_x}. \tag{20}$$





For a value of $\nu = 1$, the gamma distribution reduces to the inverse exponential or Marshall-Palmer distribution (Marshall and Palmer, 1948) in which case $N_{0x}$ (with dimensions $\mathrm{m}^{-4}$) is commonly referred to as the intercept parameter. The different categories are $x = \{\text{ash, cloud droplet, raindrop, ice crystal, graupel}\}$. In addition to having different sizes and phase, raindrops and graupel are considered to be sedimenting, and graupel has a higher density than ice.

Most atmospheric models employ a moment-based microphysics scheme for the treatment of cloud microphysics. The main

essence of these schemes involve assuming that the PSD can be pre-described using a functional form such as the gamma distribution function and one or more of the moments of the PSD are simulated prognostically. An analytical solution for $M_p$, the $p$-th moment of the size distribution can be expressed as

$$M_x(p) = \int_0^\infty D_x^p N(D_x) dD_x = \frac{N_{Tx}}{\lambda_x^p} \frac{\Gamma(\nu_x + p)}{\Gamma(\nu_x)}. \tag{21}$$

One-moment schemes (such as that used in ATHAM to describe the microphysics) predict the specific concentration $q_x$ ($M(3)$)

of the different categories after specifying $N_{0x}$ and $\nu_x$. Then, with the definition of $M_x$ given by equation (21), $\lambda_x$ can be related to $N_{Tx}$ and $q_x$. The mass $m_x$ of a hydrometeor species is related to its diameter $D_x$ through $m_x = c_{m_x} D_x^{p_{m_x}}$, where $c_{m_x}$ and $p_{m_x}$ are constants. For simplicity, we assume a spherical shape for all ash and hydrometeors in this work which means that $c_{m_x} = \frac{\pi}{6} \rho_x$ where $\rho_x$ is the particle density and $p_{m_x} = 3$. The specific concentration $q_x$ is given by

$$
\begin{aligned}
q_x &= \int_0^\infty \frac{\rho_x}{\rho} \frac{\pi}{6} D_x^3 N(D_x) dD_x, \\
\qquad &= \frac{\pi}{6} \frac{\rho_x}{\rho} \frac{N_{0x}}{\lambda_x^{\nu_x + 3}} \Gamma(\nu_x + 3).
\end{aligned}
\tag{22}
$$

The slope parameter $\lambda_x$ can then be derived as

$$\lambda_x = \left( \frac{\pi}{6} \frac{\rho_x}{\rho} \frac{N_{0x}}{q_x} \Gamma(\nu_x + 3) \right)^{1/(\nu_x + 3)}. \tag{23}$$

The radar reflectivity module in SynRad starts with calculating $N_{0x}$ or $\lambda_x$ from $q_x$ using equation (23). For the hydrometeors, a size distribution $N_{Dx}$ to be used in equation (11) is then generated from (19) using known constant values of $N_{0x}$ and $\nu_x$.

For the ash particles, $N0_{\text{ash}}$ is calculated by re-arranging equation (22) and values of $\lambda_{\text{ash}}$ and $\nu_{\text{ash}}$ retrieved from the initial size distribution and described in section 4.1.1.

### 3.4 Calculation of complex permittivity

At microwave frequencies, the dielectric properties of air, water and ice can be expressed using the complex permittivityCalculation $\epsilon = \epsilon' + i\epsilon''$ where the the first term is the real part of the permittivity and the imaginary part $\epsilon''$ is the loss factor. The

complex refractive index can also be represented in a similar form as $m = m' + im''$. The relation $m = \sqrt{\epsilon}$ is used to relate the two quantities.





The dielectric constant of water is usually modeled using the Debye formula (Liebe et al., 1991) given by

$$\epsilon' = \epsilon_\infty + [\epsilon_s - \epsilon_\infty]/[1 + (\lambda_s/\lambda)^2] \tag{24}$$

$$\epsilon'' = [\epsilon_s - \epsilon_\infty(\lambda_s/\lambda)]/[1 + (\lambda_s/\lambda)^2] \tag{25}$$

where $\epsilon_\infty$ is the high frequency dielectric constant ($f \to \infty$), $\epsilon_s$ the static constant and $\lambda_s$ the relaxation wavelength (Ray, 1972).

The model proposed in Hufford (1991) is used to calculate the dielectric constant of ice where

$$\epsilon' = 3.15, \quad \epsilon'' = \alpha(T)/f + \beta(T)f$$

with $T$ the temperature in °C. The coefficients $\alpha$ (GHz$^{-1}$) and $\beta$ (GHz$^{-1}$) are given by

$$\alpha = (50.4 + 62\theta) \times 10^{-4} e^{-22.1\theta},$$

$$\beta = \left(\frac{0.502 - 0.131\theta}{1 + \theta}\right) \times 10^{-4} + 0.542 \times 10^{-6} \left(\frac{1 + \theta}{\theta + 0.0073}\right)^2,$$

$$\theta = \frac{300}{273.15 + T} - 1,$$

The complex dielectric constant of graupel is calculated by considering graupel as a mixture of ice and air (Ryzhkov et al.,
2011). The final form of the equation can be represented in the Debye form as

$$\frac{\epsilon_g - 1}{\epsilon_g + 2} = \frac{\rho_g}{\rho_i} \frac{\epsilon_i - 1}{\epsilon_i + 2}, \tag{26}$$

where $\rho_g = 700 \, \mathrm{kg \, m^{-3}}$ and $\rho_i = 917 \, \mathrm{kg \, m^{-3}}$ are the densities of graupel and ice respectively.

## 4 Application

### 4.1 Case study - Raikoke 2019 eruption

The eruption of the Raikoke volcano on 21 June 2019 is selected to demonstrate the working of SynRad v1.0. Raikoke is located at 48.2°N, 153.3°E and is on an uninhabited, small ($4.6 \, \mathrm{km^2}$) volcanic island situated near the middle of the Kuril Island arc in the Sea of Okhotsk in the northwest Pacific ocean. There are no monitoring stations with microwave weather radars to track the eruption and existing studies have been through satellite data. The eruption lasted 3.5 h and the ash cloud rose to a height of 13 km. Due to the lack of field studies, the ash size distribution from the 2011 Grimsvotn eruption is adopted
for numerically modelling the Raikoke eruption using ATHAM. The initial sizes and specific concentrations of the initial ash distribution are given in table 2. So for the purposes of evaluating SynRad, we place a C-band weather radar south of the vent in the numerical domain. The radar has the same characteristics as the EEC MM-250C Doppler meteorological radar used in Schneider and Hoblitt (2013) to monitor the Redoubt eruption in Alaska. The characteristics required as input to SynRad are given in table 1. The MDS is calculated for this radar system by using a minimum detectable dBZ of 0 dBZ.







**Figure 2.** Equivalent radar reflectivities $Z_e$ calculated using the Mie theory of the total scattering cloud (ash + hydrometeors) every five minutes post eruption. The figure shows $Z_e$ at a vertical cross-section across the vent.

| Type | C-band |
|---|---|
| Wavelength | 5 cm |
| Peak transmitted power | 245.2 kW |
| Beam width | 0.9° |
| Elevation angle | 0.5° |
| Pulse duration | 2.15 µs |
| Actual gain of antenna | 44.9 dB |
| Minimum Detectable Signal | -93.9 dBm |

**Table 1.** Specifications of the weather radar for which synthetic signals are generated using SynRad.





### 4.1.1 Retrieving the parameters of the ash size distribution

Assuming a gamma distribution, the initial ash sizes are binned into eight different ash categories with different radii and densities as given in table 2. It should be noted that specific concentrations are cited with units $\mathrm{kg\,kg^{-1}}$ in table 2 as this is the standard unit for the specific concentration prognostic variable in ATHAM (mass per unit kilogram of total mixture). For the rest of the manuscript, the concentrations are cited in $\mathrm{g\,m^{-3}}$ or $\mathrm{kg\,m^{-3}}$ by multiplying the specific concentrations with the density of the mixture (gas + hydrometeors). Since no known measurements exist for the Raikoke eruption, we consider data from the Grimsvotn 2011 eruption which had similar characteristics. This data was downloaded from the Independent Volcanic Eruption Source Parameter Archive (IVESPA) Version 1.0, (ivespa.co.uk, (Aubry et al., 2021)).The parameters of the gamma PSD are then obtained by fitting the gamma PSD function given by equation (19) to the values in table 2. Rather than fitting to the number concentration of the ash particles, the function is fit to the volume density, i.e., the volume of ash particles present per size bin. This is done by considering the diameters of the eight different ash categories to be the middle diameter in each size bin. Using the initial specific concentrations given in table 2, the volume densities are calculated and a function of the form given by equation (19) is fit to this data using the scipy optimize.curve-fit function in python. The three-parameters of the gamma function retrieved from this calculation are : $\nu = 1.74$, $N_0 = 0.116 \, \mathrm{m^{-(\nu+3)}}$ and $\lambda = 458.07 \, \mathrm{m^{-1}}$. The fitted function is shown in figure 3 (black line) along with the initial piece-wise constant values (red dots) of the volume density of the size distribution. A fitted function such as this can result in an increased volume density of the larger particles which can usually result in an overall higher volumetric radar reflectivity. To analyze the sensitivity of the radar reflectivities to the tail of the distribution, we assumed the size distribution of each individual ash category to be described by an individual gamma function over the size ranges specified in column 4 of table 2. The total PSD is then represented by the superposition of the PSD of the eight categories. Figure 4 shows this comparison where a higher $Z_e$ is seen when representing the size distribution by a single gamma function. The white spaces in sub plot (a) appear because only grid cells with individual ash category greater than a threshold is selected whereas for (b) the threshold is applied on the total ash concentration. Even though there is an obvious higher $Z_e$ with this method, this reduces computation time significantly (as the alternate method has to do the same calculation eight times) and we opt to represent the ash size distribution using one single gamma function as shown in figure 3.

The radar reflectivity module is used to calculate $Z_e$ at different times post eruption and shown in figure 2 as a cross-section across the vent. The plume reaches a maximum height of almost 20 km within 15 min. We choose the results at $t = 15$ min to showcase the ability of SynRad to capture the characteristics of the volcanic plume in the following section.

### 4.2 Results

Three key outputs from the radar reflectivity module are the equivalent radar reflectivities $Z_e$, the vertical maximum intensity (VMI) or the composite reflectivity, which is calculated as the maximum in $Z_e$ (dBZ) in each vertical column, and the radar echotop which is the maximum cloud top height in each vertical column. These fields are chosen since these are the commonly seen operational radar images. For SynRad calculations, we only consider grid points with total specific concentrations (of ash and hyrdrometeors) greater than $10^{-8}$ kgkg$^{-1}$. This is to reduce the computational time as signals from range cells with





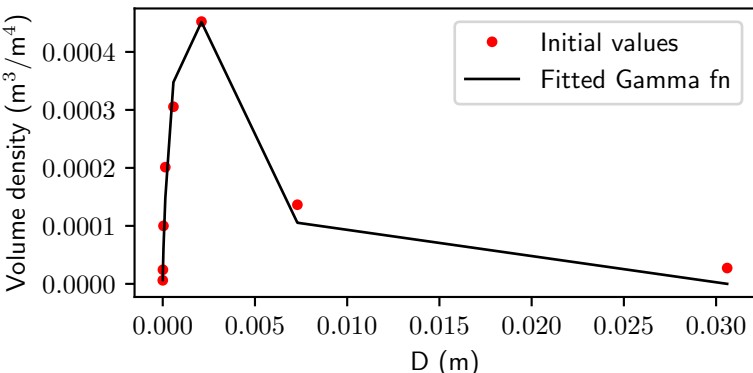

**Figure 3.** Initial volume density (the volume of ash particles for each size bin) for ash (red dots). The black line shows the fit to the volume density.

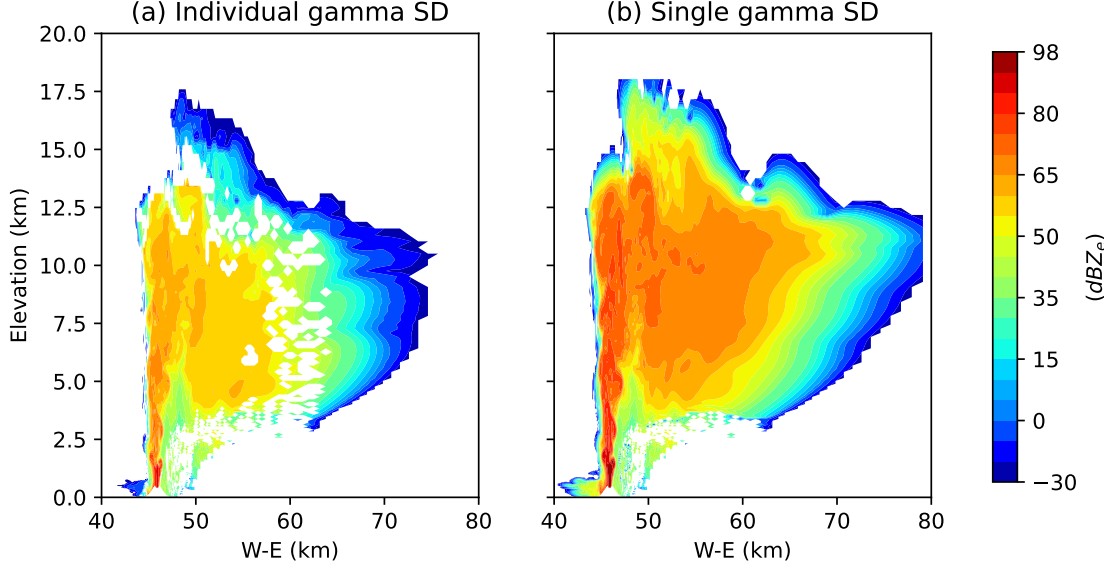

**Figure 4.** Equivalent radar reflectivities at a W-E cross-section at the vent at $t = 15$ mins post-eruption. The two plots show $Z_e$ with (a) the superposition of separate gamma distribution for each ash category and (b) one single gamma function to represent the entire size distribution of ash.





| Category number | Category label | Diameter | Bin range | Density ($\mathrm{kg\,m^{-3}}$) | Initial concentration ($\mathrm{kg\,kg^{-1}}$) | Size parameter |
|---|---|---|---|---|---|---|
| 1 | Ultra fine ash | $2\,\mu\mathrm{m}$ | $1.75\,\mu\mathrm{m}$ - $6\,\mu\mathrm{m}$ | 2300 | 0.01 | $1.26\times10^{-4}$ |
| 2 | Fine ash | $10\,\mu\mathrm{m}$ | $6\,\mu\mathrm{m}$ - $25\,\mu\mathrm{m}$ | 2300 | 0.04 | $6.28\times10^{-4}$ |
| 3 | Small ash | $40\,\mu\mathrm{m}$ | $25\,\mu\mathrm{m}$ - $90\,\mu\mathrm{m}$ | 2100 | 0.15 | $2.5\times10^{-3}$ |
| 4 | Coarse ash | $140\,\mu\mathrm{m}$ | $90\,\mu\mathrm{m}$ - $0.104\,\mathrm{mm}$ | 1600 | 0.23 | $8.8\times10^{-3}$ |
| 5 | Large ash | $0.6\,\mathrm{mm}$ | $0.104\,\mathrm{mm}$ - $1.43\,\mathrm{mm}$ | 1100 | 0.24 | $3.6\times10^{-2}$ |
| 6 | Fine lapilli | $2.2\,\mathrm{mm}$ | $1.43\,\mathrm{mm}$ - $4.7\,\mathrm{mm}$ | 650 | 0.21 | 0.132 |
| 7 | Coarse lapilli | $7.2\,\mathrm{mm}$ | $4.7\,\mathrm{mm}$ - $1.7\,\mathrm{cm}$ | 513 | 0.05 | 0.46 |
| 8 | Bomb | $3.0\,\mathrm{cm}$ | $1.7\,\mathrm{cm}$ - $3.5\,\mathrm{cm}$ | 513.0 | 0.01 | 1.92 |

**Table 2.** Properties of the initial grain size distribution for the Raikoke eruption modelled using ATHAM. The size parameter $x = \pi D/\lambda$.

concentrations below this threshold is guaranteed to be below the MDS and hence not detected by the radar. Once the return signal $P_r$ is calculated for each of these grid points, a threshold (equal to the MDS for a corresponding radar) is applied on the

$P_r$ field to remove all return signals less than the MDS. Then, the maximum value of $Z_e$ and the cloud top in each of these vertical columns gives the SynRad obtained VMI and the echotop, respectively.

In this section, these fields from SynRad will be compared with the ATHAM output. For this, ATHAM output is first converted to comparable fields. Figure 5 shows the (a) maximum plume heights and (c) VMIs calculated from ATHAM output at $t = 15$ minutes. To calculate the maximum plume heights, a threshold of $2 \times 10^{-4}\mathrm{gm^{-3}}$ is applied to the total concentration

(ash + hydrometeors) fields in ATHAM and the highest point in each vertical column with concentrations above this threshold is identified. This threshold is chosen as this is the concentration limit at which 'enhanced procedures' are to be implemented by airlines due to the danger the ash poses to the aircraft jet engines. colorblue To calculate the ATHAM VMIs, the specific concentrations are converted to equivalent radar reflectivities $Z_e$ using equation (16). The VMI is then the maximum of these dBZ$_e$ values in each vertical columns.

At $t = 15$ minutes, SynRad captures most of the ash cloud as shown in figure 5. Following the meteorological conditions of the day, the ash cloud is dispersed to the east and spreads to the north with cloud heights varying from a maximum of approximately 20 km closer to the vent to approximately 10 kms in the distal ash cloud at x = 80 km and y = 30 km. The cloud echo tops or the radar indicated cloud top near the vent detected by the C-band radar as calculated by SynRad (figure 5b) are less than the maximum plume heights of the ATHAM cloud (figure 5a). Using the ash concentrations (in $\mathrm{kg\,m^{-3}}$) of

the individual categories, we calculated the maximum plume heights of each ash category and the results are shown in figure 6. This analysis revealed that the first four ash categories ($D = 2\,\mu\mathrm{m}$ - $140\,\mu\mathrm{m}$, figures 6a-d) dominate the maximum plume height field in figure 5a. The maximum plume heights for the larger four ash categories (figures 6e-h) are well below those of the finer ash clouds. The C-band radar cannot detect the fine ash cloud (the first four categories in this case) especially close to the vent (orange and red cloud) and the distal cloud (light green cloud). This could be because of the attenuation due to the

presence of larger ash (near the vent) in the signal path or the lower concentrations leading to a weaker return signal (in the distal cloud) from the smaller four categories of ash . This will be explored in detail later in the section.



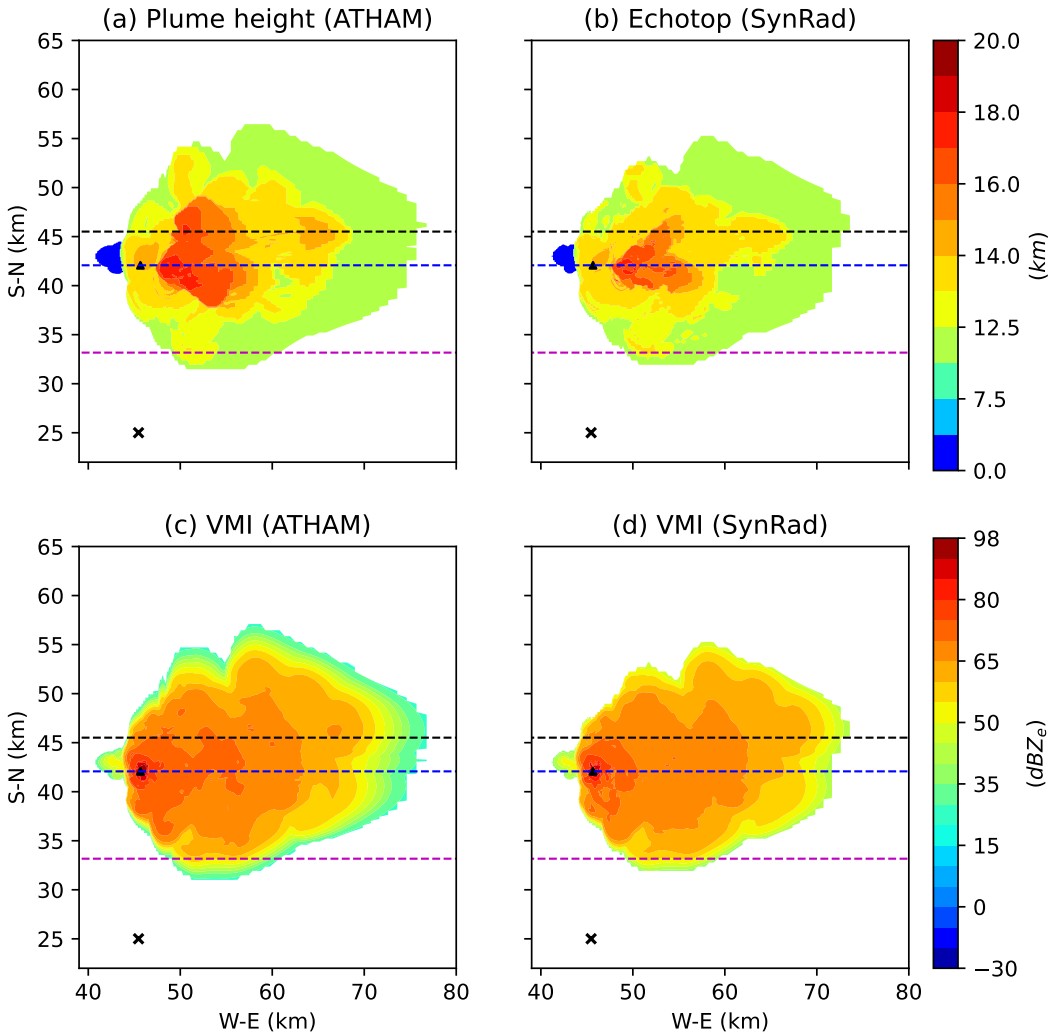

**Figure 5.** Comparison of ATHAM and SynRad results. The fields under comparison are (a) ATHAM plume top height calculated from simulated tracer concentrations, (b) SynRad echotop, (c) ATHAM maximum radar reflectivites in a vertical column, and (d) SynRad VMI at t=15 mins. The vent location is represented by the triangle and the cross denotes the position of the radar. The ATHAM plume heights are calculated as the highest point in a vertical column with total concentration greater than or equal to $2 \times 10^{-4}$ g/m$^3$ and SynRad echotops are the highest point in a vertical column from which a return signal is detected. The dashed-lines represent the three different W-E cross-sections where the results will be shown in the following figures. These are locations before the vent (pink dashed line), at the vent (blue dashed line) and behind the vent (black dashed line).







**Figure 6.** Plume heights calculated from ATHAM data at t = 15mins for individual ash categories by applying a threshold of $2 \times 10^{-4} \text{g/m}^3$ on individual ash concentrations.



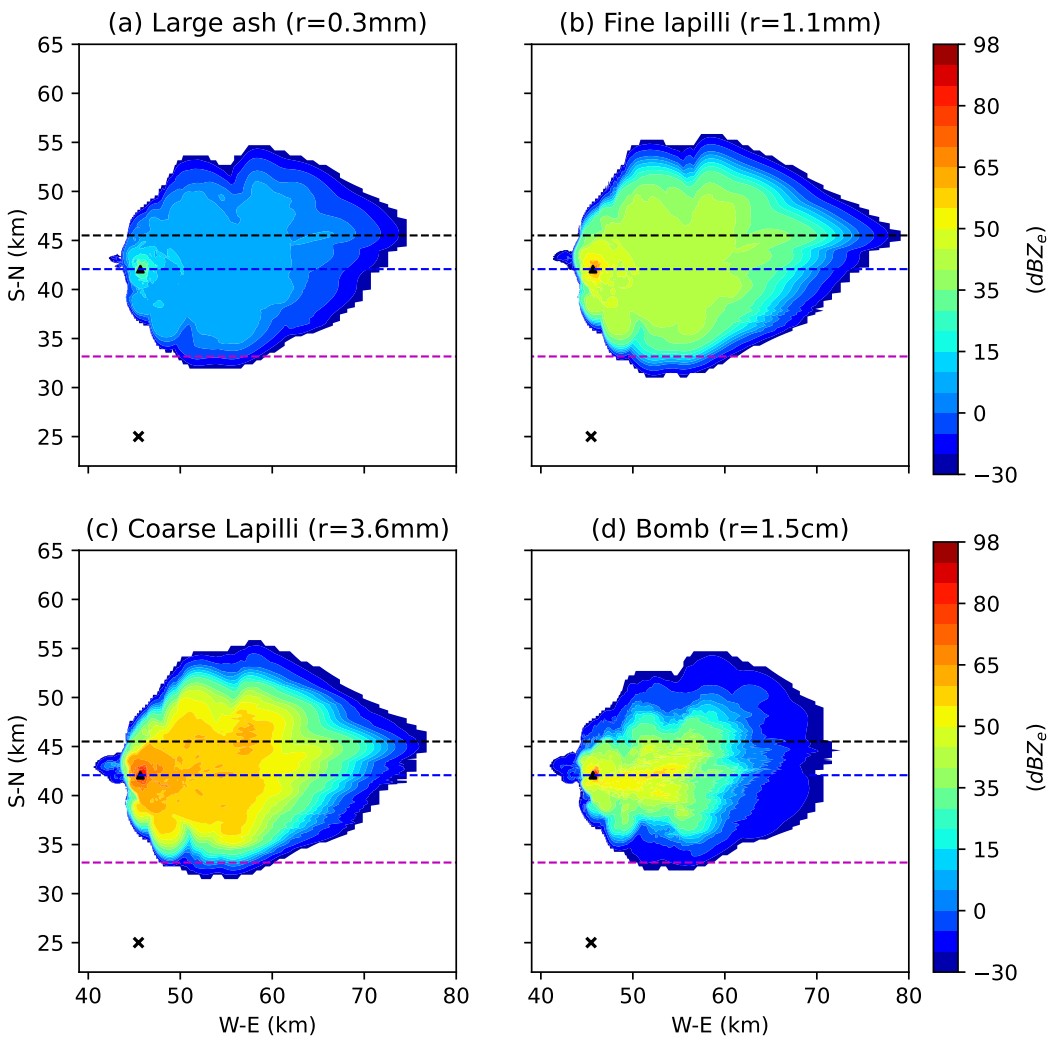

**Figure 7.** VMI fields calculated from ATHAM data at t = 15mins for (the four largest) individual ash categories. The VMIs for the smaller four categories (not shown) are well below the -30 dBZ threshold imposed in this study. The dashed lines are the same as in figure 5.



The VMIs are a measure of the reflectivities of the larger and highly scattering particles and there is a very good agreement between figures 5(c) and 5(d) especially closer to the vent (and even in the distal cloud). The VMIs shown in figure 5(c) is completely dominated by the ash particles as the maximum reflectivities of the hydrometeors are always less than that of the

larger ash particles. Even within ash, certain categories are detected preferentially by the radar. The reflectivity contribution from total ash concentrations in one cell is calculated by assuming the size distributions to follow one single gamma distribution (as detailed in section 4.1.1). An analysis to calculate individual ash category contributions to the VMI field was performed by numerically splitting the ash size distribution to 8 different size ranges (with each size in table 2 assumed as the center of the bin). This revealed that the first four ash categories ($D = 2\,\mu m$ - $140\,\mu m$) have maximum reflectivities well below -30

dBZ and hence do not have a significant contribution to the total ash volumetric reflectivity. As discussed earlier, this means that these categories are not detected by the radar. On the other hand, the relatively large wavelength of the C-band radar also means that they are likely to preferentially detect the high reflectivity tephra (i.e. the larger ash categories) and hydrometeors, and this ultimately decides the VMI values. Comparing figure 5c to the VMIs of the larger ash categories (figure 7), it is likely that the radar preferentially detects the coarse lapilli. In the Rayleigh regime, there is an almost linear increase in the particle

radar cross-sections with diameters which then plateaus once the Mie regime is reached. For a particular ash category to lie in the Rayleigh or Mie regime, the size parameter $x = 2\pi r/\lambda \ll 1$. The values of this parameter given in table 2 reveals that the last ash category (bombs) fall in the Mie regime. This explains why the coarse lapilli have a stronger return signal compared to the bombs which are larger in size. At the distal cloud, the C-band radar fails to detect the ash particles (shades of light green in figure 5d) which dominate the maximum reflectivities at these distances (as is also apparent in the echo tops) in figure 5.

At these distances, the four larger ash categories are usually present in smaller concentrations with the VMIs dominated by the finer ash cloud (which is undetected) leading to the difference between the two figures at these low reflectivities. Also, the radar fails to capture the high reflectivity region east of the vent (50 - 60 kms along the x-axis) in figure 5(c). As will be shown in figure 8, the radar signal is attenuated resulting in a weaker signal in this region.

The path attenuation (from the radar location to the vent) experienced by the C-band radar due to the ash and hydrometeors

is investigated next in figure 8 at three cross-sections (represented in figure 5 as dashed lines). Figures 8a - c are cross-sections corresponding to the red, blue and black dashed lines in figure 5 respectively. These cross-sections are selected to study the effect of attenuation at different locations relative to the vent and radar location. The first line (red) corresponds to a location $9\,km$ in front of the vent where we expect only finer ash and hence lower attenuation. The second (blue) is at the vent where we expect larger tephra and the third line (black) is a location $3.4\,km$ behind the vent where the rays will have had to pass

through a significant amount of attenuating ash cloud. The color bars range from 0 to 1 indicating a non-attenuated and fully attenuated signal respectively. Figure 8(a) shows zero to minimal path attenuation. At this cross-section, even though the radar rays are travelling through all the ash categories, the reflectivities of the four bigger ash categories are well below zero which in the C-band results in no or negligible attenuation. At a cross-section across the vent 8(b), heavy attenuation is seen vertically above and close to the vent. This is the attenuation experienced by the radar ray when it travels through the small region around

the vent with very large and coarse tephra. The category-wise split of the VMIs reveals that the region of high reflectivity most responsible for the attenuation is actually the coarse lapilli (category 7) as they are present in a higher concentration than the



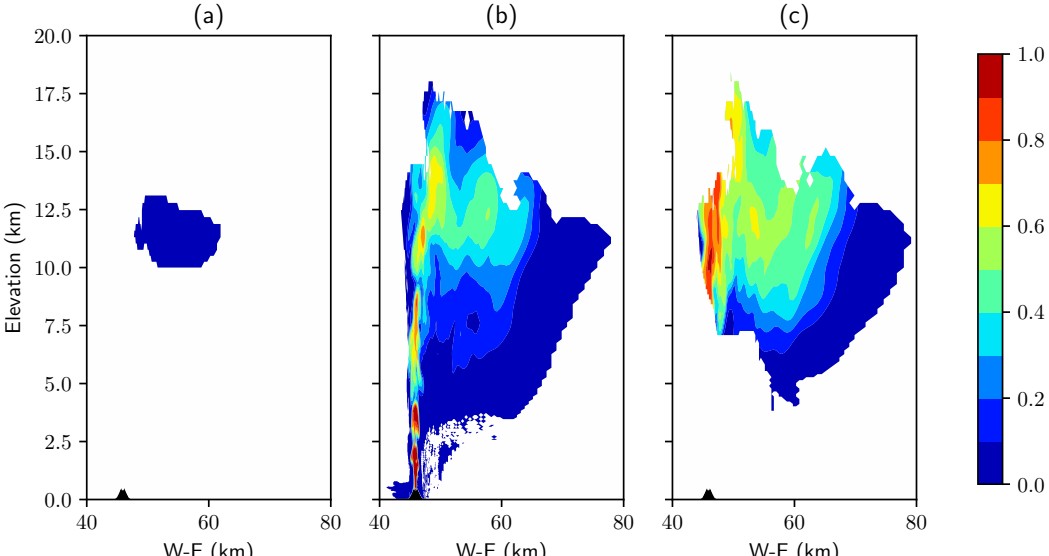

**Figure 8.** Signal attenuation calculated by the ray tracing module of SynRad at three different W-E cross-sections in the numerical domain at $t = 15$ mins- (a) before the vent (pink dashed line in figure 5), (b) at the vent (blue dashed line in figure 5) and (c) behind the vent (black dashed lines) in figure 5.

larger bombs (category 8). This is the dark red patch around the vent in figure 5(c) with reflectivities above 80 dBZ. This patch of tephra with high reflectivities is also responsible for the high extinction of the signal (slender red column) in figure 8(c). Cross-section in between these two locations reveal a similar pattern (not shown) with the slender red region stretching from the vent to altitudes up to 15 kms as in figure 8c. The broader attenuation experienced by the signal up to 65 km east in figures (b) and (c) is from when the beam passes through the high reflectivity region (> 60 dBZ shown in orange/dark yellow).

The return power $P_r$ (dBm) from each radar gate is presented in figure 9. The grey halo around the ash clouds represent the actual cloud from ATHAM data. These are all points with total specific concentrations greater than $10^{-5}$ g/kg, i.e. all points where we are calculating return signals from as mentioned at the start of the section. Most of the ash cloud in figure 9a is made up of high reflectivity tephra and hydrometeors as is evident in figure 10a where the minimum reflectivity in the cloud is around 20 dBZ. This combined with the negligible path attenuation experienced by the beam up to this cross-section and the relative closeness of the radar to the target volume results in a return signal that is above the MDS of the C-band radar. Figure 9b however reveals a counter intuitive picture. As the signal undergoes a degree of path attenuation as revealed by figure 8b, we expect a weak signal from the slender column experiencing the heavy attenuation and the return power to reflect the attenuation cross-section. However, we see a stronger signal than expected in the slender column. This is a combination of two facts: one, even though the signal is heavily attenuated, it is not fully extinguished, i.e. , it does not drop to zero. Secondly, the slender column has the largest in size and the high reflectivity tephra which results in a high return signal even when multiplied with the attenuation value (as in equation (7)). This results in the return power signal cross-sections closely mirroring the cross-



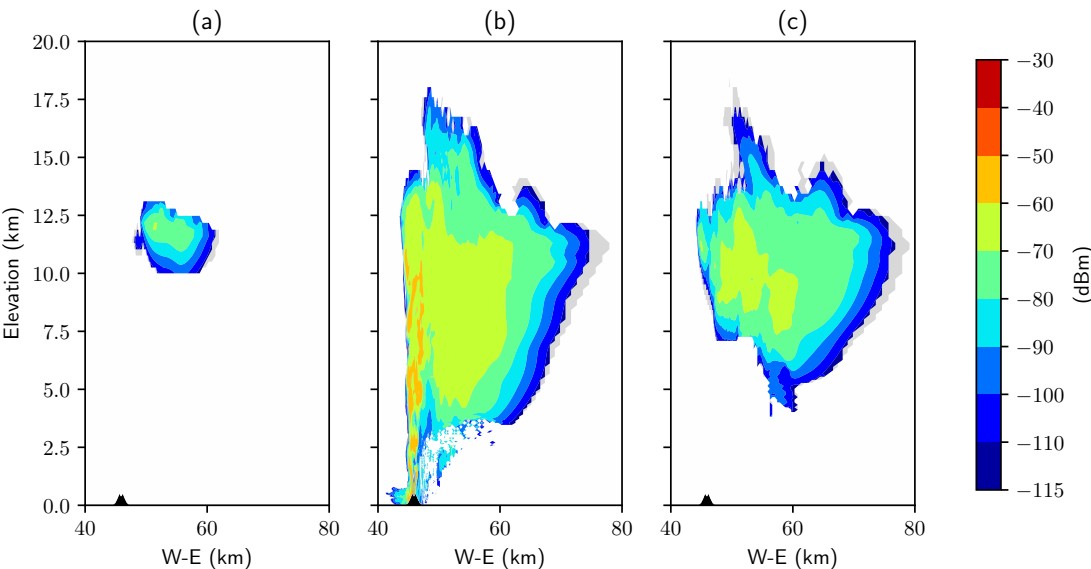

**Figure 9.** The return signal power $P_r$ in dBm calculated by SynRad at three different W-E cross-sections - (a) before the vent (pink dashed line in figure 5), (b) at the vent (blue dashed line in figure 5) and (c) behind the vent (black dashed lines) in figure 5.

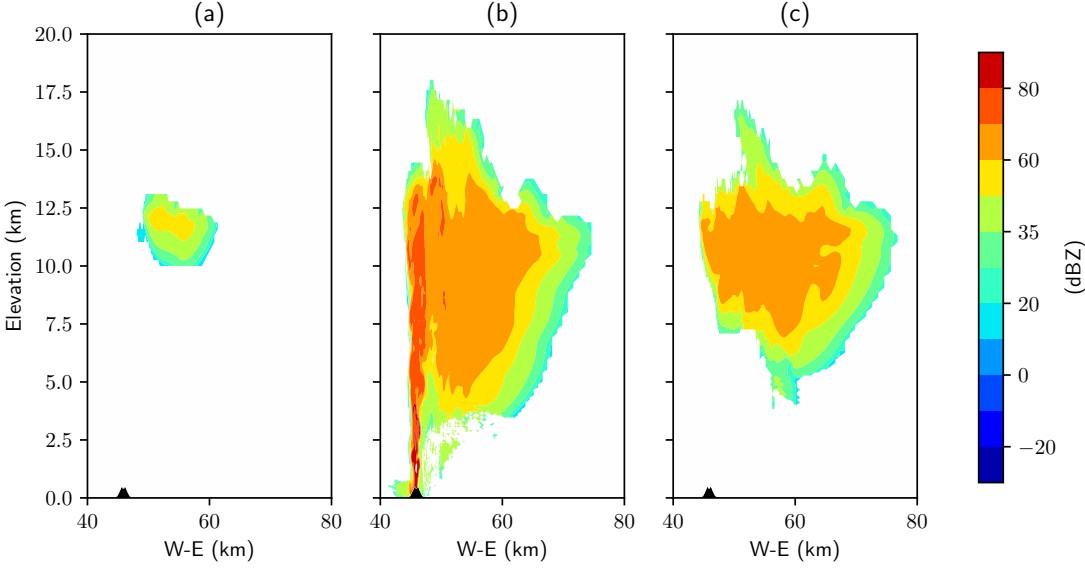

**Figure 10.** Equivalent radar reflectivities (dBZ) calculated by SynRad at three different W-E cross-sections - (a) before the vent (red dashed line in figure 5), (b) at the vent (blue dashed line in figure 5) and (c) behind the vent (black dashed lines) in figure 5.





sections for the equivalent radar reflectivities $Z_e$ (figure 10, move to appendix) rather than the cross-sections for the attenuation.
We believe that for an eruption which results in tephra of smaller size categories or for a radar of higher frequencies (where the signal is completely attenuated to zero), the return power signal cross-sections will closely mirror the attenuation cross-sections. This theme is followed in figure 9c as well with the cross-section further behind the vent revealing a return signal at odds with the attenuation cross-section. However, another interesting feature in this figure is the more obvious appearance of the grey halo suggesting that this frequency band misses out on capturing the outer bands of the ash cloud. Comparison with
the attenuation figure suggests that this is not due to the signal being attenuated but rather due to the range, which leads to the return signal at the radar from these distances being lower than the MDS.

## 5 Conclusions

A recently developed radar forward operator, SynRad, which simulates the radar measurement process and generates synthetic radar signals from three-dimensional output from volcanic plume models is presented. It simulates key radar observables
like equivalent radar reflectivity, path attenuation, return signals, vertical maximum intensities, and echotops. The operator is specially designed to simulate the radar measurement processes behind the detection of volcanic clouds where attenuation due to the volcanic ash is key to deciding how deep into the ash cloud the signal can penetrate. Also considered is the scattering or attenuation due to hydrometeors such as cloud droplets, raindrops, ice crystals and graupel. SynRad produces return signals from each grid point in the three-dimensional output that it is working on. A key feature of this operator is the ray tracing
module which tracks radar beams from the antenna to the cloud (and back) and calculates the attenuation along its path. The high computational cost of the ray tracing module leads us to opt for certain simplifications such as assuming the radar beam to propagate as a single ray to each grid cell (rather than assuming the actual volume of the beam). We also ignore the Doppler observations, beam bending and assume the radar to be single polarized.

The 2019 eruption of the Raikoke volcano is used as the case study to showcase the capabilities of SynRad. The measurement
process of a C-band radar is simulated and the equivalent reflectivities, VMIs and echotops are calculated. The operator was able to generate realistic fields of the equivalent radar reflectivities and the echotops and VMIs from SynRad show a good agreement with those generated from ATHAM.

SynRad can be used to identify the best possible locations for operationally deploying radar to monitor the ash cloud post eruption. Locations can be identified which allows maximum penetration into and hence detection of the ash cloud. Another
key application is that SynRad can be used to identify the optimal frequency or frequencies (for a multi-frequency radar) which balances the trade-off between a higher return signal and the higher path attenuation that comes at these higher frequencies. In addition, the synthetic radar signals can be used to develop and test radar retrieval algorithm for the retrieval of ash properties including their mass and size. SynRad can also be used in the context of verifying the cloud microphysics implemented in any plume model. This can be done by comparing the radar observables calculated from the prognostic cloud model variables and
directly comparing this with weather radar observables. The operator can also be used to assimilate radar data into prediction models. An estimate of the source parameters (such as ash concentrations and size) can be arrived at by combining a prior





forecast state with the observations/radar observables from the operator, which can then be used as an initial state of the VATDMs.

*Code availability.* The current version of the model will be made publicly available on GitHub after the manuscript is accepted for publica-
tion. The exact version of SynRad used to produce the results used in this paper is archived on Zenodo, as are input data and scripts to run the model and produce the plots for all the simulations presented in this paper. This can be accessed using the DOI: 10.5281/zenodo.11863012.

*Author contributions.* AM designed and implemented the first version of the forward operator. VN validated the operator and performed the case study detailed in this work, and prepared the manuscript. MH provided the original idea for SynRad, contributed to the design and discussions, and provided overall supervision. DM and DR contributed to the design and discussion of SynRad and provided expertise on the
radar measurement process.

*Competing interests.* The authors declare that no competing interests are present.

*Acknowledgements.* VN, MH, DM and DR acknowledge funding from NERC for the Radar-supported Next-Generation Forecasting of Volcanic Ash Hazard (R4Ash) project (NE/S004386/1 and NE/S003622/1). AM acknowledges funding from the Department of Geography at the University of Cambridge for a summer internship.





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
