# Peer review of "SynRad v1.0: A radar forward operator to simulate synthetic weather radar observations from volcanic ash clouds."

_EGUsphere, 2024_

## Referee Comment (RC2)

[referee-annotated manuscript omitted]

---

## Author Comment (AC1)

AC:

- The title could be specified by using the expression "weather radar". This manuscript exclusively addresses weather radar measurements, but volcanic clouds are also measured using other non-weather radar systems. The latter systems not necessarily produce the radar observables listed here. This distinction between weather (scanning) radars and other non-weather (profiling) radars should be clear also throughout the introduction. A good review on the observation of volcanic ash clouds using (any) radar system is given by Hort and Scharff (Hort, M. & Scharff, L.; Detection of Airborne Volcanic Ash Using Radar; In Book: Volcanic Ash; 2016; doi: 10.1016/b978-0-08-100405-0.00013-6)
- We have edited title and abstract to include 'weather' radars. The distinction is also made now in multiple places in the introduction and conclusions.

- L12: higher frequency microwave radars (K-band and higher) that observe volcanic activity do already exist. However, they may not operate in scanning mode. Please specify again: weather radar
- The following line has been added: 'Even though higher frequency microwave weather radars (K-band and higher) have been used to observe volcanic activity, they may not operate in scanning mode.' (L12-13).

- L22-25: again this describes scanning radars. Profiling radars will have a much better temporal and even spatial resolution, but only in one single direction.

  This has been corrected to specify that we are talking about scanning weather radars. (L27)

- L38: Another observation of volcanic activity using dual-polarization weather radar was done in New Zealand (Crouch, J. F.; Pardo, N. & Miller, C. A.; Dual polarisation C-band weather radar imagery of the 6 August 2012 Te Maari Eruption, Mount Tongariro, New Zealand; 2014; doi: 10.1016/j.jvolgeores.2014.05.003)
- The e ruption details have been added. (L44-45).

- L42: volcanic particles can have any size near the vent. Their size will decrease with the distance from the vent. The formulation used in the manuscript is correct ("particle sizes are smaller than raindrops especially far from the vent"), but it must also be clear that this assumption becomes invalid near the vent.
- The following lines have been added: 'Volcanic particles can have any size near the vent, but the ash particle sizes far away from the vent are smaller than raindrops. This means that frequencies higher than those used in weather radars have possible benefits while monitoring and measuring volcanic ash concentrations.' L48-51.

- L62-63: the listing could be extended by the paper by Scharff et al (2012) mentioned above. With this reference you would acknowledge the old model without explicitly mentioning it.
- The paper has been added. L71.

- L115-119: neglecting attenuation due to water vapor in the atmosphere might be feasible, but water vapor is highly abundant in the eruption cloud. Is it considered there? Effects of volcanic SO2 can be neglected at common radar-relevant frequencies but may become more important in higher frequency radars. Please mention up to what frequency this assumption is valid.
- We haven't considered water vapour within the cloud. We agree that it is highly abundant in the eruption cloud. The reason behind not including water vapour was because we envisage the attenuation within the eruption cloud especially early in the eruption to be dominated by the larger ash particles. SynRad is to be complemented with a new stand-alone package/module which includes attenuation due to all gases (in the atmosphere and within the cloud) but the result of this work is reserved for a follow-on paper.
- We haven't done a quantification of the attenuation due to SO2 currently. But with the new package we should be able to quantify this. Currently we agree that water vapour will have the dominant effect with respect to attenuation and SO2 can be neglected at least up to the W-band.

- L125: When polarization is included in the future, do you also reevaluate assumption 1 on the sphericity on particles?
- Yes, the assumptions would need to be re-evaluated. We would need to do a sensitivity study to investigate the importance of the different individual parameters for the polarimetric variables. Hopefully this might reveal dependencies that could be used to simplify relations or validate assumptions. We would need to look at the variability in density, shape and settling properties, especially of ice.

- L168-179: This paragraph does not make clear whether a one-way or two-way attenuation is calculated. From the formulation it appears as if only the attenuation from the radar to each cell is calculated.
- This was an error in the manuscript but done correctly in the code. The exponential in equation 8 has been corrected to exp(2*int_0_r \kappa dr). Additionally, a line has been added after equation 6 to clarify that L^2 is two-way path attenuation. (L155)

- L214: what is "m"? This parameter appears before its explanation in line 219.
- The explanation has been moved forward to L224.

- L337-339: Please elaborate more on the way the ATHAM VMIs are calculated. To me the difference in calculation to SynRad did not become clear enough. This leads to follow-up questions in paragraphs that include comparisons between the two VMIs (e.g. in L427). What do you expect to see from ATHAM VMIs in comparison to SynRad VMIs. Is it simply the inclusion/absence of attenuation?
- Exactly. The difference is simply the attenuation. This has been clarified with the following sentences: 'The ATHAM VMIs are calculated at each grid point and hence do not include any effects of attenuation. In essence, the difference between the

ATHAM and SynRad values for both VMI and echotops will be due to the attenuation experienced by the radar signal in SynRad.' (L352-354).

- L407-408: I do not understand this sentence. Could you please rewrite it. I suggest at least exchanging "at odds" with the more common "in contradiction to".

  Changed to 'in contradiction to'.L423.

- L410: Isn't it a combination of range, particle size and concentration that lead to a return signal lower than the MDS?
- Indeed. I have changed this to 'due to the range and the fact that we tend to have lesser concentrations of fine (smaller in size) ash at these distances'. L426.

- L430-431: This application is not shown here. Please rephrase.
- Rephrased to highlight that this is the work of a follow-up paper. L448-449.

Other notable technical corrections:

L380: the color bar does not include white (no attenuation due to absence of cloud). see also Figure 8

All non-cloudy grid points are set as NaN which is why color bar does not include white to denote absence of clouds.

Figure 8: please indicate the vertical exaggeration in the caption.

I am not sure I understand what this comment means, especially 'the vertical exaggeration'. Could you please clarify?

L401-403: Something appears wrong with this sentence. You may split this into two sentences for easier reading.

Edited. L415-417.

---

## Author Comment (AC2)

[revised manuscript text omitted]

---

## Author Response (AR3)

The following comments are for the authors' reference to improve the manuscript.

L73-74: "Even though conventional forward operators calculate the attenuated or corrected reflectivity, they do not trace the radar beam and its extinction through the cloud under study. A key aspect of SynRad is a ray tracing module that calculates the extinction of the transmitted (and reflected) radar signal due to interactions within the ash cloud." Some of the forward radar operators you mentioned do not calculate the extinction of radar signal, but they output the specific attenuation coefficient of each radar gate. With this output, users can easily obtain the path attenuation (or extinction) by accumulating their effects.

Agreed. This point has been added to manuscript. The distinction is made between SynRad and other operators by highlighting that SynRad calculates extinction along the beam **online** while it can be calculated **offline** from the specific attenuation coefficients in the other operators.

We would also like to bring the authors' attention to a new forward radar operator recently published in GMD (Xie et al., GMD, 17, 5657–5688, 2024).

This reference has now been added (L71).

L128: Why is that the atmospheric refraction is totally neglected? A simple static 4/3RE effective radius model would significantly improve the accuracy of beam trajectory calculations. Such incorporation does not introduce complexity but improves the accuracy

It is indeed due to the introduction of complexity that we decided to neglect the effect of atmospheric refraction in the first version of SynRad. The way we staged the development of SynRad was to have a simple ray tracing module initially. With respect to the trajectory calculation, this involved treating the beam as single line rays (modelling the beam-axis) from the radar location to the model grid cell centre while neglecting atmospheric refraction and hence beam bending. The second stage (which we are currently working on) will be to include the effects of refraction as well as attenuation due to the atmosphere and thereafter beam broadening.

Currently, the way we trace rays are by considering the cell centre where the radar is located as the start point of a beam. The different rays are then obtained by considering each of the cell centres of the ATHAM output grid as the end points. By considering the ray as a straight line, we then divide this into equal ray sub-segments and then interpolate values from the (non-equidistant) grid cell centres to each of the sub-segment centres. The attenuation is then calculated using the Beer-Lambert law **along the beam direction**. If we were to implement the 4/3ERM now, we would obtain a new set of latitudes and longitudes for corresponding radar elevations and azimuths by applying the method, and values would need to be then interpolated from the model grid to this new lat-lon grid. We would then divide these rays into sub segments to calculate along the beam attenuation which probably wouldn't be the attenuation experienced by the actual beam. In this way, the 4/3 ERM would be an inconvenient implementation currently and more importantly wouldn't fit in with our current along-the-beam attenuation calculation module.

Instead, an along-the-beam model for ray propagation with beam bending like the TORE (TOtal Reflection) method (Zeng et al 2014) is planned to be implemented as the next step.

This would be better suited for our attenuation calculation as well. As the main purpose of the tool is to evaluate which radar frequency or combination of frequencies would be ideal for capturing the propagation of the entirety of an ash cloud, the attenuation due to ash along the ray path was a higher priority at this stage of the development of SynRad.

*Zeng Y, Blahak U, Neuper M, Jerger D. 2014. Radar beam tracing methods based on atmospheric refractive index. J. Atmos. Ocean. Technol. 31:2650–2670*

L130: Volcanic ash and hydrometeors are currently assumed as spheres. The non-sphericity effect has not been discussed at all. If dual-polarization capability is introduced, non-sphericity and orientation preference must be considered. Otherwise, the polarimetric radar variables will be zero.

This was answered in a previous comment from a reviewer and is repeated below:

Yes, the assumptions regarding sphericity would need to be re-evaluated in the case of introducing dual polarization. We would need to do a sensitivity study to investigate the importance of the different individual parameters for the polarimetric variables. Hopefully this might reveal dependencies that could be used to simplify relations or validate assumptions. We would need to look at the variability in density, shape and settling properties, especially of ice.

We have now included the following line in the manuscript as well (L 135).

*This would also include considering non-sphericity and orientation preferences for the ash and hydrometeors.*

L320: For the single gamma size distribution method. SynRad operator must perform fit for each radar gate, is it right? I wonder whether the volume densities of different size bins match gamma distribution so well for each radar gate. I suggest authors should validate this point.

We perform a fit to a gamma distribution using the initial concentrations to obtain a value for the three different parameters: $\nu_{ash}, N_{0,ash}, \lambda_{ash}$. Then for each radar gate/grid cell, we prescribe $\lambda_{ash}$ and $\nu_{ash}$ to be fixed and calculate $N_{0,ash}$ from the prognostic ash concentration variables in each grid cell (using equation 22) and then calculate the ash number concentrations (assuming a gamma PSD). This does introduce an overestimation due to the tail of the gamma PSD. This discussion is shown in figure 4 and the corresponding text (L325-333).

4.2 While this paper presents the development and methodology of SynRad, validation against real observations has not been conducted or discussed. To strengthen the credibility of SynRad, at least one case study comparing simulation results with actual observations should be included. If there is difficulty, the missing comparison should be explained.

This manuscript is intended as an introduction of this model with the focus on concept, methodology and details. The magnitudes of the radar measurables and other results for the eruption presented in this manuscript are realistic and we feel that this gives sufficient confidence on the validity of this code. Additionally, since the Raikoke eruption was never

observed with ground-based radars and never modelled before, this is the first time this eruption has been studied in such detail. We will be submitting a comparison with actual observations as a follow-on publication in due course where detailed comparisons with radar observations and with different radar frequencies will be presented for a known and well-studied/observed eruption. Including this here would be beyond the scope of this paper.

Report #1

Two technical corrections: in line 44, please add the citation for the Tongariro eruption paper (Crouch, J. F.; Pardo, N. & Miller, C. A.; Dual polarisation C-band weather radar imagery of the 6 August 2012 Te Maari Eruption, Mount Tongariro, New Zealand; J Volcanol Geotherm Res, Elsevier BV, 2014, 286, 415-436, doi: 10.1016/j.jvolgeores.2014.05.003)

Added (L45)

and in line 368 the last word should be 'are'.

Corrected